# Comparison between USPIOs and SPIOs for Multimodal Imaging of Extracellular Vesicles Extracted from Adipose Tissue-Derived Adult Stem Cells

**DOI:** 10.3390/ijms25179701

**Published:** 2024-09-07

**Authors:** Arnaud M. Capuzzo, Giusi Piccolantonio, Alessandro Negri, Pietro Bontempi, Maria A. Lacavalla, Manuela Malatesta, Ilaria Scambi, Raffaella Mariotti, Kerstin Lüdtke-Buzug, Mauro Corsi, Pasquina Marzola

**Affiliations:** 1Department of Diagnostics and Public Health, University of Verona, Strada le Grazie, 8, 37134 Verona, Italy; arnaudmartino.capuzzo@univr.it (A.M.C.); alessandro.negri@univr.it (A.N.); 2Department of Engineering for Innovation Medicine, University of Verona, Strada le Grazie, 15, 37134 Verona, Italy; giusi.piccolantonio@univr.it (G.P.); pietro.bontempi@univr.it (P.B.); mariaassunta.lacavalla@unipd.it (M.A.L.); 3Department of Chemical Science, University of Padova, Via Marzolo 1, 35131 Padova, Italy; 4Department of Neuroscience, Biomedicine and Movement Sciences, University of Verona, Piazzale L.A. Scuro 10, 37134 Verona, Italy; manuela.malatesta@univr.it (M.M.); ilaria.scambi@univr.it (I.S.); raffaella.mariotti@univr.it (R.M.); 5Institute of Medical Engineering, University of Luebeck, Ratzeburger Allee 160, 23562 Lübeck, Germany; kerstin.luedtkebuzug@uni-luebeck.de; 6Fraunhofer Research Institution for Individualized and Cell-Based Medical Engineering IMTE, 23562 Lübeck, Germany; 7Evotec Consultant, Via A. Fleming 4, 37135 Verona, Italy; mac9967@googlemail.com

**Keywords:** MRI, regenerative medicine, iron nanoparticles, nanomedicine, extracellular vesicles

## Abstract

Adipose tissue-derived adult stem (ADAS) cells and extracellular vesicle (EV) therapy offer promising avenues for treating neurodegenerative diseases due to their accessibility and potential for autologous cell transplantation. However, the clinical application of ADAS cells or EVs is limited by the challenge of precisely identifying them in specific regions of interest. This study compares two superparamagnetic iron oxide nanoparticles, differing mainly in size, to determine their efficacy for allowing non-invasive ADAS tracking via MRI/MPI and indirect labeling of EVs. We compared a USPIO (about 5 nm) with an SPIO (Resovist^®^, about 70 nm). A physicochemical characterization of nanoparticles was conducted using DLS, TEM, MRI, and MPI. ADAS cells were labeled with the two nanoparticles, and their viability was assessed via MTT assay. MRI detected labeled cells, while TEM and Prussian Blue staining were employed to confirm cell uptake. The results revealed that Resovist^®^ exhibited higher transversal relaxivity value than USPIO and, consequently, allows for detection with higher sensitivity by MRI. A 200 µgFe/mL concentration was identified as optimal for ADAS labeling. MPI detected only Resovist^®^. The findings suggest that Resovist^®^ may offer enhanced detection of ADAS cells and EVs, making it suitable for multimodal imaging. Preliminary results obtained by extracting EVs from ADAS cells labeled with Resovist^®^ indicate that EVs retain the nanoparticles, paving the way to an efficient and multimodal detection of EVs.

## 1. Introduction

In 2001, mesenchymal stem cells (MSCs) from adipose tissue were shown to be adipose tissue-derived adult stem (ADAS) cells [1]. For patients suffering from various neurodegenerative diseases, stem cells are emerging as a promising therapeutic approach to neuronal replacement and regeneration [2]. Due to their widespread availability, propensity to move to injured tissue, capacity to differentiate, and ability to aid in the healing process in various neurodegenerative diseases, adipose stem cells have attracted significant interest for possible therapeutic uses [3].

An additional advancement in the development of therapeutic agents based on ADAS cells is represented by the potential use of extracellular vesicles (EVs) extracted from them. Indeed, ADAS cells appear to have a paracrine effect that arises from released EVs [4]. EVs are small vesicles (30–200 nm) related to the type of secreting cell, containing proteins, lipids, and nucleic acids. In an in vitro model of amyotrophic lateral sclerosis (ALS), results obtained in our institution suggest that EVs enhance the repair of damaged areas by releasing their contents. This highlights the potential of EVs as a novel cell-free therapeutic approach, offering a safer alternative to cell-based treatments [5].

Even though stem cell-mediated therapy is a novel and promising approach to treating a wide range of neurodegenerative diseases, certain issues still need to be addressed, such as determining how precisely cells home to the target. In this respect, magnetic resonance imaging (MRI) has emerged as a non-invasive tool for cell tracking. However, to be detected via MRI, it is necessary to label the stem cells with superparamagnetic iron oxide nanoparticles (SPIONs), which act as contrast agents. These contrast agents are widely employed for in vivo cell tracking, demonstrating that MRI of labeled cells is a useful method for assessing cellular migration toward the lesion site [6,7,8,9,10,11]. These iron-based superparamagnetic nanoparticles (NPs) operate as negative contrast agents, primarily affecting signal intensity by shortening T_2_ and T_2_ * relaxation times, thereby producing a dark area of hypo intensity in T_2_ images and a signal loss in the area surrounding the iron in T_2_ * images (a so-called “blooming effect”) [12].

MRI is a very useful tool in this field because it is easily accessible, non-invasive, and radiation-free. It provides high spatial resolution and contrast, allowing for a detailed visualization of soft tissues. This resolution and contrast are essential for detecting small concentrations of labeled cells within biological tissue. Thanks to noninvasiveness, MRI can be used to acquire real-time or time-sequence images, allowing for dynamic monitoring of the movement and behaviour of labeled cells. This capability is particularly useful for studying biological processes, such as targeting labeled cells to specific tissues or their release in response to certain stimuli. However, MRI has also some limitations, which include an inability to detect labeled cells in tissues with low signals, such as lungs or haemorrhagic regions. In this context, integrating MRI with another imaging modality could be highly beneficial in overcoming the inherent limitations of MRI. Magnetic Particle Imaging (MPI) is an emerging non-invasive imaging technology that directly detects SPIONs, offering high specificity and sensitivity for in vivo tracking and quantifying SPION-labeled cells. Central to MPI’s functionality is the field-free region (FFR), a zone within the gradient selection field where SPIONs can respond to an oscillating excitation field, enabling their detection and localization. Utilizing alternating excitation magnetic fields and pick-up coils, MPI captures the spatial distribution of SPIONs, which is crucial for various applications, including cell tracking, vascular imaging, oncology, pulmonary imaging, inflammation assessment, and magnetic hyperthermia. The ability to produce real-time, high-quality images with exceptional positive contrast, even in deep tissues, positions MPI as a promising diagnostic tool with vast potential in medical imaging [13,14]. It is useful for both the ex vivo and in vivo visualization and quantification of iron oxide-based contrast agents. MPI provides dynamic and quantitative imaging of the concentration of magnetic NPs, in contrast to a traditional MRI [15,16].

As mentioned above, to be detected in MRI and MPI, stem cells must be labeled with NPs, thereby causing a distinguishable change in signal strength. Superparamagnetic NPs have been used frequently to magnetically label stem cells in this regard [17]. It should be noted that superparamagnetic NPs have been proven to be safe and are commonly used in medical procedures like MRI [18]. MRI tracking remains a valuable tool in regenerative medicine, as it enables the monitoring of disease evolution in pathologies that require stem cell therapy [15,16].

Previously, our group has published some papers on labeling with superparamagnetic NPs and in vivo visualization of different types of cells by MRI [19,20]. Other research groups have also reported similar findings, showing efficient labeling, marginal cytotoxicity, and sensitive detection of cells labeled with SPIONs [21,22,23].

Our group has proposed an innovative method to label EVs, which subsequent literature has referred to as the indirect method. This process involves first labeling the parent stem cells and then extracting the EVs. After extraction, these vesicles naturally contain the label. In our initial study [20], limiting to a single modality imaging, we chose an ultrasmall SPIO (USPIO) with a hydrodynamic diameter of about 5 nm. The idea was to use NPs smaller than EVs to marginally perturb their structure and functionality. Of note, these USPIOs had no detectable signal in MPI even at the highest concentration available, corresponding to 5 mgFe/mL [13]. Therefore, it is necessary to test larger iron oxide NPs to evaluate their ability to label parent cells and the EVs extracted from them without negatively affecting their functionality. We directed our research to Ferucarbotran Resovist^®^, a SPION contrast agent that is currently used in clinics, which is also commercially known as VivoTrax™ [24] or Cliavist^®^ [25] for experimental studies.

It is crucial that MRI and MPI are sensitive to the same tracers. Therefore, labeling stem cells and EVs with SPION contrast agents will, in principle, allow for a multimodal imaging approach for investigating stem cells and EVs homing in vivo. This approach will help overcome the limitations of the two imaging techniques and exploit their synergistic potential.

The first part of this study aims to evaluate the physical–chemical characterization of Resovist^®^ in comparison with USPIO. The second section aims to establish the dose of Resovist^®^ and USPIO compatible with cell viability culture and detectability, in vitro and ex vivo, of ADAS cells in MRI. Finally, in the third part of the study, we extracted labeled EVs from ADAS cells, previously labeled with Resovist^®^, and we briefly analysed their characteristics. This work represents a comparison between USPIO and SPIO labeling of ADAS cells to determine which NP offers better feedback values for tracking stem cells and EVs using non-invasive techniques such as MRI and MPI.

## 2. Results

### 2.1. Nanoparticle Characterization

The physico–chemical properties of USPIO and Resovist^®^ are reported in Table 1 and Figure 1. Representative dynamic light scattering (DLS) histograms are included in the Appendix A. There was a substantial difference between the hydrodinamic size of USPIO (9.5 ± 2.01 nm) and Resovist^®^ (69.18 ± 1.56 nm). The two NPs also differed in their zeta potential, with Resovist^®^ measuring −31.8 ± 7.12 mV, slightly more negative than USPIO, which measured −23.6 ± 1.7 mV. Transmission electron microscopy (TEM) images, reported in Figure 1B, confirmed the actual dimensions of the NPs, 5.00 ± 0.01 nm for USPIO and 70.0 ± 0.2 nm for Resovist^®^, which are in agreement with the datasheets provided and the literature data [26,27]. To better illustrate the structure of the NPs used, high-resolution TEM images are included in the Appendix A.

The ability of NPs to shorten the relaxation times of water was obtained by the measurement of longitudinal and transversal relaxivities in agarose gel phantoms (Figure 1A). As expected, the transversal relaxivity r_2_ of Resovist^®^ was higher than that of USPIO (274.9 ± 4.1 mM^−1^ s^−1^ vs. 103.3 ± 2.1 mM^−1^ s^−1^). The longitudinal relaxivity was very low for both NPs, resulting in a very high value of the r_2_/r_1_ ratio, which is a favorable characteristic for T_2_-relaxing contrast agents.

To assess the detection limit of Resovist^®^ in MPI, 12 phantoms were prepared with decreasing Fe concentrations, ranging from 200 µg to 1 µg of Fe in 150 µL of distilled water. The results are presented in Figure 1C. The upper part shows a linear correlation between the Fe concentration and the signal intensity. In the bottom part, representative MPI images showing signal intensities of Resovist^®^ are reported. Within the used experimental setup, the lowest detectable concentration of Resovist^®^ was determined to be 1 µgFe/150 µL. This can be observed in the Appendix A, where images normalized to the maximum intensity signal of each image are provided for each concentration. The MPI signal was not detectable for USPIO even at the highest tested dosage corresponding to the mother solution (5 mgFe/mL).

### 2.2. Toxicity and Internalization of NPs in ADAS

Internalization of NPs in ADAS cells was obtained by incubation of NPs in the culture medium. An MTT assay was used to assess the potential cytotoxicity of both NPs. The data presented in Figure 2A reveal a mild, concentration-dependent cellular toxicity associated with the NPs, which also varies with the incubation time. Specifically, it was observed that, after 24 h of incubation, neither USPIO nor Resovist^®^ showed statistically significant differences compared to the control ADAS cells (CTRL). However, after 48 h of incubation, a small, although statistically significant, difference (*p* < 0.01) was observed between ADAS cells labeled with USPIO at a concentration of 200 µgFe/mL and CTRL. Therefore, MTT data showed that both NPs (up to 200 µgFe/mL) could be used for cell labeling. No statistical differences were observed in the case of ADAS cells labeled with Resovist^®^, indicating that different concentrations of Resovist^®^ do not interfere with the viability of the cells [28]. Prussian Blue staining was conducted to assess the internalization of NPs by ADAS cells after 24 h (Figure 2B, first row) and 48 h (Figure 2B, second row) of incubation time. The reduction of ferric to ferrous iron resulted in the formation of blue precipitates, which served as an indicator of NP presence. Staining of labeled ADAS cells revealed blue areas, indicating NPs, with typical perinuclear localization. The increase in iron concentration and incubation time led to more intense Prussian Blue staining, indicative of increased NP internalization. Furthermore, at equivalent concentrations and time points, ADAS cells labeled with Resovist^®^ demonstrated greater internalization compared to those labeled with USPIO, suggesting the advantage in using larger NPs. Regardless of the NP type, localization in both cases was primarily perinuclear and widespread throughout the cytoplasmic extensions. This aspect is very favorable as it corresponds to the area of EV formation.

To evaluate the detectability and the detection limit in MRI, ADAS cells previously labeled with both NPs were immobilized in different gel tubes (agarose at 4% wt/wt) and used as phantoms. Tubes containing unlabeled cells in agarose gel were used as controls. Figure 3 shows the results for ADAS cells labeled with USPIO (upper row) and with Resovist^®^ (lower row). Various quantities of cells (10, 50, 100, 1 × 10^3^, 1 × 10^4^, 5 × 10^4^, and 1 × 10^5^) labeled using different Fe concentrations (200, 50, and 6.2 µgFe/mL) were analyzed after 24 h (Figure 3A and Figure 3C for USPIO and Resovist^®^, respectively) and 48 h (Figure 3B and Figure 3D for USPIO and Resovist^®^, respectively) of incubation time. In both cases, the findings highlighted the efficacy of MRI in pinpointing cells labeled with NPs. Indeed, these appeared as hypointense spots, indicating the presence of iron. In both cases, the ADAS cells labeled with the highest Fe concentration and incubated for 48 h exhibited enhanced detectability. A visual comparison between the two cases clearly demonstrated the superior detectability of ADAS cells labeled with Resovist^®^. Specifically, under optimal conditions, i.e., ADAS cells labeled with 200 µgFe/mL and incubated for 48 h, the minimum number of detectable cells was 100 for those cells labeled with Resovist^®^, compared to 5 × 10^3^ for cells labeled with USPIO. Figure 3 shows that the relationship between the concentration of Fe added to the culture medium and the minimum number of cells detected by MRI is not linear. For example, when the SPION concentration is reduced from 200 µgFe/mL to 6.2 µgFe/mL—a reduction of approximately 32 times—it requires 100 times more cells to be detected. Similarly, reducing the concentration from 200 µgFe/mL to 50 µgFe/mL requires 10 times more cells instead of the expected four times more if the relationship was linear. These findings can be explained by the fact that, while increasing the concentration of NPs in the culture medium does lead to an increase in the amount of iron internalized by the cells, this increase is not necessarily linear. Moreover, MRI technique used in this context is not quantitative. This is a well-known limitation of the method as reported in the literature [29].

The ultrastructural morphology of labeled ADAS cells is reported in Figure 4. Both USPIO (Figure 4A) and Resovist^®^ (Figure 4B) entered the cells via endocytosis as NP clusters. Then, several NPs were found to be enclosed into small vacuoles occurring just below the cell surface membrane and into heterogeneous vacuolar structures corresponding to secondary lysosomes or residual bodies. No USPIO or Resovist^®^ was found free in the cytosol or in the nucleus. The cytoplasmic organelles and the nuclei of cells containing NPs showed excellent morphological preservation. TEM observation demonstrated that both NPs enter the cells as clusters following the endocytic pathway and remain enclosed into vacuols from uptake to degradation. These could explain the absence of any cellular stress sign.

### 2.3. Imaging with Magnetic Resonance

Having established a protocol enabling the efficient detection of ADAS cells in vitro, we conducted studies to investigate whether ADAS cells labeled with NPs could be localized by MRI in ex vivo muscle tissue. MR images of pre- and post-intramuscular hindlimb injection of 5 × 10^3^ labeled ADAS cells are shown in Figure 5. Specifically, the ADAS cells labeled with USPIO are reported on the left (Figure 5A) and those labeled with Resovist^®^ on the right (Figure 5B). Similar to phantoms, the hypointense areas correspond to labeled ADAS (red circle). Note that the size of the regions of hypointensity is related to the pulse sequence used. In fact, in the fast low-angle shot (FLASH) sequence, the labeled ADAS cells cause an alteration of the local magnetic field within a volume that exceeds the volume filled by the cells, which is a feature called the “blooming effect” [30]. It is surprising that the dark area due to the presence of cells labeled with USPIO is larger than that due to the cells labeled with Resovist^®^. This can be possibly due to the different sites where the cells were injected that could affect the area where they spread. Specifically, in the case of USPIO, they entered the posterior subcutaneous region, while Resovist^®^ entered the muscle tissue.

To evaluate the detectability and detection limit of ADAS cells labeled with Resovist^®^ in mouse brains, varying amounts of ADAS (5 × 10^3^, 1 × 10^4^, 5 × 10^4^, 1 × 10^5^, and 1 × 10^6^), labeled at a concentration of 200 µgFe/mL and incubated for 24 h, were suspended in a volume of 10 µL and injected in excised mouse brain. Figure 6A shows MRI acquired on these brain samples, highlighting the effectiveness of MRI in identifying cells labeled with Resovist^®^ (red circles). Qualitatively, the size of dark areas was proportional to the number of cells injected, with a larger number of cells resulting in a more extensive hypointense region and a wider area of signal distortion in FLASH images. ADAS cells were clearly visible even at the smallest numbers, such as 5 × 10^3^ and 1 × 10^4^, especially in the FLASH sequence where the presence of Resovist^®^ caused a disturbance of the local magnetic field. Furthermore, it is important to note that the black dots around and external to the brain are associated with air bubbles. While a significant limitation of this experiment is that it was conducted ex vivo, we believe that visualizing cells in excised mouse brains remains crucial, as the target organ of the proposed therapeutic approach is brain tissue. Additionally, we would like to clarify that the administered volume was 10 µL, which is relatively large for a mouse brain. This may account for the observed cell leakage from the injection site, a phenomenon that, although not ideal, is understandable due to the high injection volume. Nevertheless, Figure 6 demonstrates that despite these technical challenges, cells are detectable within the target tissue, underscoring the relevance of our approach.

Finally, to confirm that what is observed in Figure 6A is associated with iron and not with lesions from the injection site, an in-depth analysis with histological Prussian Blue staining (Figure 6B) was performed.

### 2.4. Extraction of Extracellular Vesicles from ADAS-Labeled Cells

In a previous study [20], our group demonstrated that EVs derived from ADAS labeled with USPIO retained the labeling. In this paper, we investigate the use of larger NPs, specifically Resovist^®^, to label ADAS cells. While results previously reported demonstrated the capability of Resovist^®^ to label parent cells, it is still necessary to establish whether the EVs obtained from Resovist^®^-labeled ADAS cells retain the labeling. EVs were isolated, following established protocol [2,4,5,19,20,31], from labeled ADAS supernatants obtained after 24 h of incubation with 200 μgFe/mL of Resovist^®^.

Figure 7A shows nanoparticle tracking analysis (NTA) measurements obtained from EVs extracted from unlabeled cells (on the left) and from those labeled with Resovist^®^ (on the right). EVs extracted from labeled cells showed a slight size increase in the population around 100 nm. In both cases, the mode was very similar, indicating that there were no drastic increases in the size between the two samples. EVs were quantified using protein concentration BCA assay, yielding 290.648 μg/mL of exosomal proteins for control unlabeled EVs and 319.229 μg/mL of exosomal proteins for EVs labeled with Resovist^®^.

Figure 7B shows T_2_ map images of unlabeled EVs (EV CTRL) marked with a red square, the Resovist^®^-labeled EVs marked with a red circle, a 1:10 solution of Resovist^®^-labeled EVs marked with a red pentagon, and water not marked. It can be observed that the labeled EVs are slightly darker than those not labeled. In Figure 7C, the T_2_ relaxation time curves of EVs CTRL, labeled EVs, 1:10 solution of labeled EVs, and water are reported. The curves for the water (T_2_ time = 824 ± 1.79 ms) and EV CTRL (T_2_ time = 863 ± 0.44 ms) are almost overlapped, indicating that the EV CTRL does not alter the relaxation time as expected. In contrast, the labeled EVs (T_2_ time = 528 ± 0.47 ms) show a faster decay in T_2_ relaxation time due to the presence of Resovist^®^. Even in the 1:10 solution of EVs labeled with Resovist^®^ (T_2_ time = 753 ± 0.34 ms), a slight decrease in T_2_ relaxation time is noted.

## 3. Discussion

ADAS cells are regarded with increasing interest as a potential therapeutic approach in different pathologies. The abundance of adipose tissue in the body provides a convenient and minimally invasive source for harvesting these cells, making them an attractive option for various therapeutic applications. Over the years, numerous investigations have demonstrated that, rather than integration and survival in host tissues, the paracrine activity of ADAS may be the source of their therapeutic potential. EVs naturally carry bioactive molecules, which can help modulate immune responses and promote tissue repair and regeneration. The capability to visualize and track stem cells and their products, like EVs, with a noninvasive technique such as MRI, is crucial in understanding the underlying biological mechanisms for any potential therapeutic approach. Superparamagnetic NPs can be employed as contrast agents for MRI to affect the local magnetic field, proton relaxation times, and signal intensity. Previous research conducted by our group demonstrated that ADAS cells and EVs, labeled with USPIOs, can be effectively tracked both in vitro and in vivo using MRI [20]. Building on our previous research, we decided to test SPIOs, specifically Resovist^®^, due to their higher relaxivity, clinical approval, and the fact that they can also be detected using MPI. The hypothesis of this study is that Resovist^®^, having a larger size and relaxivity than USPIO, is more suitable for labeling parent cells. To this purpose, we compared the established labeling method based on USPIOs with a similar protocol based on larger iron oxide NPs, like Resovist^®^, that is an optimal candidate for labeling: well-known, stable, biocompatible, and sufficiently small to be incorporated in cells and EVs [32,33,34].

Physical and chemical characterization of NPs is required prior to biological application. One of the initial analyses involved measuring the hydrodynamic size of the NPs using DLS and TEM. Differently from SPIOs, DLS results revealed aggregation phenomena in the case of USPIO, a finding visually confirmed by TEM. This observation suggests that Resovist^®^ may be more suitable for labeling purposes. MRI characterization demonstrated that both NPs acted as negative contrast agents and were suitable for T_2_ and T_2_ * imaging, but the transversal relaxivity of Resovist^®^ is higher than that of USPIO. Resovist^®^ has a transverse relaxivity more than twice that of USPIO, indicating that Resovist^®^ causes a faster decay of MRI signals and offers higher contrast in T_2_-weighted images. This aspect is crucial for optimal sensitivity in tracking and detecting labeled ADAS cells [35]. Finally, Resovist^®^ was detectable via MPI. This outcome was expected, as Resovist^®^ has a larger core diameter compared to USPIO, resulting in a stronger MPI signal [26].

Cytotoxicity is a critical aspect to consider when using NPs for cell labeling. Thus, the two NPs were compared to assess how their internalization affected ADAS cell viability. ADAS cells remain viable even at the highest concentration tested when labeled with Resovist^®^. In contrast, a small but statistically significant decrease in viability is observed between control cells and cells labeled with 200 µgFe/mL of USPIO after 48 h of incubation time. This outcome suggests that it is possible to use Resovist^®^, even at an effective dose for MRI, without significantly compromising cell viability. In a previous study conducted by our team with USPIO [20], cytotoxicity was assessed using the Trypan Blue assay, which did not reveal significant differences between the control and treated groups. However, in the present study, the MTT assay revealed a small difference between the control group and the USPIO-labeled cells at 48 h. While the Trypan Blue assay did not reveal any significant differences between the control group and the cells labeled with USPIO at 48 h, the MTT assay did, indicating that it may be more sensitive and accurate in identifying changes in cell viability. The Trypan Blue assay is based on the dye’s capability to cross the membrane of dead cells only. Therefore, non-viable cells appear blue, while live cells remain unstained. The MTT assay evaluates the reduction of the tetrazolium dye MTT to purple formazan (which is quantified colorimetrically) by mitochondrial dehydrogenases, thus providing an estimate of living cells. Therefore, the MTT assay may be more sensitive in detecting even small variations in the vital activity of the cells [36].

Furthermore, Prussian Blue images at comparable Fe concentrations showed that cells labeled with Resovist^®^ exhibited a more pronounced blue coloration. Considering the interaction between larger NPs and cells, our results suggest that Resovist^®^ is more suitable for experimental use. TEM images showed that labeled ADAS cells incorporate Resovist^®^, as well as USPIO, through endocytic mechanisms. As previously reported in the literature [11,19], NPs accumulate inside ADAS and their multivesicular bodies via endocytosis, which may then fuse with the plasma membrane and release labeled EVs.

MRI experiments were designed to determine the lowest detectable numbers of NP-labeled cells and the highest NP concentrations compatible with cell viability. At the highest concentration used, 200 µgFe/mL, our results showed that MRI can detect up to 1 × 10^3^ ADAS cells labeled with Resovist^®^ after 24 h and 100 cells after 48 h of incubation time. Similarly, MRI can detect up to 1 × 10^4^ ADAS cells labeled with USPIO after 24 h of incubation and 5 × 10^3^ after 48 h. In our previous work [20], we were able to detect as few as 100 cells labeled with USPIO. However, it is important to note that the experimental conditions differed from those in the current study. In the previous work, we employed a 72 h incubation period, whereas in the present study, we focused on the earlier time points of 24 h and 48 h. Moreover, USPIO has been shown to be internalized more effectively with longer incubation times.

Furthermore, we successfully confirmed the detection of ADAS cells labeled with NPs in MRI following intramuscular injection into the hindlimb muscle. MRI effectively identified up to 5 × 10^3^ cells labeled with either NPs. Post-intramuscular MRI images revealed that ADAS cells labeled with Resovist^®^ appeared more localized in the hindlimb muscle, likely due to the injection site. In contrast, ADAS cells labeled with USPIO were found in an inguinal subcutaneous location, a mouse adipose depot, which increases the dispersion of ADAS cells. In both cases, the injections were successful, resulting in a hypointense signal spot attributable to the presence of the NPs. An interesting finding from these data is that the 5 × 10^3^ ADAS cells labeled with the NPs incubated for 48 h are well identifiable in both agarose phantoms and in skeletal muscle tissue.

Considering the above-mentioned advantages of using the larger NPs—namely improved detection in MRI even at low cell counts, higher r_2_ relaxivity for better negative contrast, visibility in MPI, lower cell toxicity, and superior internalization in ADAS cells after 24 h of incubation—we decided to label ADAS cells with Resovist^®^ for proof-of-concept imaging in the brain tissue. We, therefore, labeled ADAS with Resovist^®^ at a concentration of 200 µFe/mL and locally injected them into excised and fixed mouse brains. Subsequently, we performed MRI to determine the minimum number of detectable labeled cells. The results demonstrated the successful detection of Resovist^®^-labeled cells at high cell counts. A comparison of histological Prussian Blue staining and MRI data revealed that hypointense areas visualized in FLASH images could significantly exceed the actual volume occupied by the injected cells. This phenomenon, known as the “blooming effect”, is characterized by a non-linear increase in the hypointense zone as the iron content increases [37]. At low cell counts, i.e., 5 × 10^3^ cells, it can be noted that in the T_2_ images, the presence of labeled ADAS cells was not clearly detected, whereas in the FLASH images, a hypointense spot was observed. Histological analysis confirmed the presence of Resovist^®^ with Prussian Blue staining. This suggests that even with a small number of labeled cells, the FLASH sequence is suited for detecting the presence of NPs where T_2_ imaging fails. However, it is important to acknowledge that the administration was conducted in excised brain tissue using an insulin syringe, which may not be ideally suited to contain the required volume of 10 µL. As a result, it is possible that some of the volume containing the labeled ADAS cells may have leaked out, potentially influencing the detectability of the labeled ADAS cells in situ. In contrast, intramuscular injection of 5 × 10^3^ labeled cells into the hindlimb muscle resulted in clear identification of the labeled ADAS in MRI images. Considering that the mouse brain has a higher T_2_ relaxation time compared to the muscle, the relative reduction of the signal intensity due to the presence of Resovist^®^ could be more pronounced in the in vivo brain scenario. Despite this, the findings suggest the potential for in vivo cell tracking, enabling precise identification of labeled cells in specific regions of interest.

The results presented in Figure 7 demonstrate the successful labeling of EVs with Resovist^®^. NTA showed a slight size increase in the population around 100 nm of EVs labeling with Resovist^®^. The mode particle size remains consistent at around 93 nm for both labeled and unlabeled EVs, suggesting that the NPs effectively attach to or incorporate into the EVs without significantly altering their size. This result is somewhat surprising since the 70 nm Resovist^®^ particles have been, at least to a certain extent, incorporated into EVs. It remains to be elucidated whether this result can be due to modifications of EVs that could, in principle, affect the therapeutic efficacy. The T_2_ relaxation map and the T_2_ relaxation curves provide further confirmation of effective labeling. The T_2_ relaxation curves offer quantitative support, revealing differences in the relaxation times of labeled EVs compared to unlabeled EVs. These findings are significant for several reasons. First, the labeling of EVs with Resovist^®^ does not significantly alter their size profile. This is crucial for ensuring that labeled EVs can still perform their native roles while being tracked or imaged. Second, the enhanced T_2_ relaxation rates of the labeled EVs underscore their potential as effective MRI contrast agents.

Based on our results, it is suggested that Resovist^®^ may be more advantageous compared to USPIO. The larger size of SPIONs, like Resovist^®^, not only improves MRI detection due to longer transverse relaxivity but also enhances histological imaging, leading to better visualization of targeted structures. Moreover, Resovist^®^ is detectable in MPI, providing high sensitivity and dual-modal capability. Developing dual-modal imaging strategies is crucial for enhancing the effectiveness of cell-based therapies and accelerating their clinical translation.

## 4. Materials and Methods

### 4.1. Physical–Chemical Characterization of Nanoparticles

Commercial USPIO (magnetite Fe_3_O_4_; Sigma–Aldrich Co., St Louis, MO, USA; catalog #725331, particle size 4–6 nm, stock solution 5 mg Fe/mL) and Resovist^®^ (magnetite Fe_3_O_4_/γ–Fe_2_O_3_; Bayer Schering Pharma GmbH, German; particle size 45–60 nm, stock solution 28 mgFe/mL) were used. Both NPs were characterized from a physical–chemical point of view using DLS and TEM. DLS measurements were performed at 25 °C using a Zetasizer Nano ZS (Malvern Instruments, Malvern, UK) operating at λ = 633 nm and equipped with a back scattering detector (173°). Each measurement was the average of six or seven data sets acquired for 10 s. DLS was used to measure the coated NPs hydrodynamic size and zeta potential. TEM was used to unveil the morphology of the structure.

MRI relaxivities of both NPs were measured using a Bruker Tomograph operating at 7 T (Bruker BioSpin, Ettlingen, Germany). The scanner was operated using Paravision, version 6.0.1 (Bruker). The longitudinal relaxation times T_1_ of aqueous solutions containing different concentrations of Fe (200, 100, 50, 25, 12.5, and 6.2 µgFe/mL, i.e., range between 0.5 mM and 0.025 mM) were measured by acquiring T_1_ maps with a RARE (rapid acquisition with relaxation enhancement) sequence, with the following parameters: TR min = 170 ms, TR max = 12,500 ms, TE = 7 ms, FOV = 5 × 6 cm^2^, MTX = 128 × 128, slice thickness = 1.5 mm, number of slices = 8, and NEX = 1. The transverse relaxation time T_2_ of the same solutions was measured by acquiring T_2_ maps with an MSME (multi-slice multi-echo) sequence with the following parameters: TR = 2000 ms, 25 equally spaced echoes, TE min = 7.5 ms, TE max = 187.5 ms, FOV = 5 × 6 cm^2^, MTX = 256 × 256, slice thickness = 1.5 mm, number of slices = 8, and NEX = 4. The signals coming from each phantom at different TEs and TRs were analyzed to obtain the relaxation curve decay, which was fitted with a single exponential function to obtain the T_2_ and T_1_ values.

MPI signal intensity was measured using a MOMENTUM MPI system (Magnetic Insight, Alameda, CA, USA). Twelve Resovist^®^ phantoms were prepared with descending Fe concentrations, ranging from 200 µg to 1 µg of Fe, in 150 µL (from 24 to 0.12 mM) for MPI signal calibration. Two-dimensional images were acquired with a 3.055 T/m selection field in a 12 cm × 6 cm FOV and drive field strength of 18 mT and 20 mT, respectively, in the X and Z axes. A custom-made MATLAB 2023a (MathWorks, Natick, MA, USA) script was used to measure the standard deviation (SD) of the system noise by scanning the sample holder multiple times with the same imaging features as for the phantoms. For each phantom, a mask image was generated, retaining the pixels with a signal equal to or greater than five times the SD (5 × SD) of the noise, according to the Rose Criterion [38], while rejecting pixels with signals below the set threshold.

### 4.2. Internalization of Nanoparticles into Stem Cells

ADAS cells were isolated from inguinal adipose tissues of 8-/12-week-old C57Bl/6 mice (Charles River, Italy), as previously described [3,4,5,20,39,40,41,42]. All mouse experiments were conducted in accordance with experimental guidelines approved by the University of Verona Committee on Animal Research (Centro Interdipartimentale di Servizio alla Ricerca Sperimentale) and by the Italian Ministry of Health (protocol #56DC9.N.BLC). In order to evaluate the cytotoxicity of both NPs, the MTT (3-(4,5-dimethylthiazol-2-yl)-2,5-diphenyltetrazolium bromide) assay was conducted. Six thousand cells/well were seeded in 96 wells with 100 µL of cell culture medium (Dulbecco’s Modified Eagle Medium 10%, Fetal bovine serum 1% and Penicillina-Streptomicina 1:1 mix). Two time points were established (24 h and 48 h) to investigate the viability of ADAS cells labeled with both NPs. NP concentration was set at 200, 100, 50, 25, 12.5, and 6.2 µgFe/mL. Absorbance reading was conducted at a spectrophotometer microplate reader with a primary filter at 570 nm and secondary at 630 nm (ChroMate; RayBiotech, Peachtree Corners, GA, USA). Cell viability data were analyzed using two-way ANOVA multiple comparisons using GraphPad Prism software, version 9 (GraphPad, San Diego, CA, USA).

MRI acquisitions were performed in agarose phantoms to assess the detectability of labeled cells. Both NPs were added to ADAS cells at concentrations of 200, 50, and 6.2 µgFe/mL. Afterward, the culture medium was aspirated, cells were washed twice with 2 mL of 1X DPBS (Dulbecco’s phosphate-buffered saline no Ca or Mg), and trypsin (0.05% trypsin prewarmed to 37 °C) was added. ADAS cells were counted, and the appropriate number of cells (10, 50, 100, 1 × 10^3^, 5 × 10^3^, 1 × 10^4^, 5 × 10^4^, and 1 × 10^5^) was placed inside a 1.5 mL Eppendorf tube with 1 mL of 1X DPBS and centrifuged at 14,000 rpm for 5 min at room temperature. Surnatant was removed. ADAS cells were resuspended in 10 µL of DPBS and injected into a gel matrix phantom (1% agarose in distilled water). To enhance the sensitivity of image contrast to changes in local magnetic susceptibility, MRI images were acquired using a FLASH sequence with TR = 1000 ms, TE = 15 ms, flip angle = 15°, FOV = 6.0 × 5.0 cm^2^, MTX = 256 × 192, slice thickness = 0.5 mm, N slice = 45, and NEX = 10.

Histological Prussian blue staining, counterstained with hematoxylin and eosin, was performed on cells to visually assess labeling. Twenty thousand ADAS cells per well were seeded onto a round 13 mm autoclaved coverslip and allowed to grow for 24 h. The cells were then incubated with 500 µL of 200, 50, and 6.2 µgFe/mL concentrations of both NPs for 24 and 48 h. The internalization of the USPIO and Resovist^®^ NPs was analyzed using an Olympus BX51 optical microscope (Olympus Italia S.r.l., Segrate, MI, Italy) equipped with a QICAM Fast 1394 Digital Camera (QImaging, Surrey, BC, Canada) for image acquisition.

In order to unveil at ultrastructural level how and where the NPs are internalized after 24 h in the ADAS, TEM was performed on the cell pellets. ADAS cell pellets were fixed in 2% glutaraldehyde in pH 7.4 Sorensen buffer for 2 h, postfixed for 2 h in 1% osmium tetroxide, dehydrated in acetone at increasing concentrations, and finally embedded in Epon–Araldite (all reagents from Electron Microscopy Sciences, Hatfield, PA, USA). One µm sections were examined under light microscopy after toluidine blue staining using an Olympus BX51 microscope (Olympus Italia S.r.l., Segrate, MI, Italy) equipped with a QICAM Fast 1394 Digital 116 Camera (QImaging, Surrey, BC, Canada) for image acquisition. Ultrathin sections (70 nm in thickness) were cut and placed on Cu/Rh grids (Reichert, Wien, Austria). TEM images were acquired with a Philips Morgagni 268 D TEM (Fei Company-Philips, Eindhoven, The Netherlands) operating at 80 kV and equipped with a Megaview G3 camera (EMSIS Gmbh, Munster, Germany) for digital image acquisition.

### 4.3. Magnetic Resonance Imaging

As a first step, to confirm the detectability of the cells labeled with both NPs, MRI was acquired after intramuscular administration in mice. Five thousand labeled cells, suspended in 100 μL of sterile DPBS, were injected ex vivo by intramuscular injection in the quadriceps of the hind leg in two mice. MRI images were acquired using a T_2_-weighted sequence (RARE) with TR = 730 ms, TE = 24 ms, FOV = 4.0 × 4.0 cm^2^, MTX = 256 × 256, slice thickness = 1 mm, N slice = 9, and NEX = 8. A T_1_-weighted FLASH sequence was also acquired with TR = 750 ms, TE = 7.5 ms, flip angle = 15°, FOV = 4.0 × 4.0 cm^2^, MTX = 192 × 192, slice thickness = 0.5 mm, N slice = 31, and NEX = 8.

To determine their detectability in the brain, labeled cells (5 × 10^3^, 1 × 10^4^, 5 × 10^4^, and 1 × 10^5^) were injected into fixed brains using a 100 µL insulin syringe, and then MRI images were acquired. High-resolution T_2_-weighted images (RARE sequence) were acquired with TR = 3000 ms, TE = 30 ms, FOV = 2.0 × 2.0 cm^2^, MTX = 192 × 192, slice thickness = 0.5 mm, N slice = 30, and NEX = 6. Moreover, a T_1_-weighted FLASH sequence was acquired with TR = 500 ms, TE = 7 ms, flip angle = 30°, FOV = 2.0 × 2.0 cm^2^, MTX = 192 × 192, slice thickness = 0.5 mm, N slice = 30, and NEX = 10. After MRI, brains were cut with a microtome, and histological Prussian blue staining was performed.

### 4.4. Extraction of Extracellular Vesicles from ADAS-Labeled Cells

EVs were isolated from 15 mL of culture medium containing 1 × 10^6^ ADAS previously incubated with 200 μgFe/mL of Resovist^®^ for 24 h. To isolate EVs from ADAS cells’ culture-conditioned medium and to avoid any contamination of shed membrane fragments and vesicles from serum, FBS deprivation for 48 h was made. Cell culture supernatants were then collected and PureExo^®^ Exosome isolation kit (101Bio, CA, USA) was used for EVs isolation, following the manufacturer’s protocol [43]. EVs were stored at 4 °C for up to 1 week. The determination of the protein content of EVs was done by Bicinchoninic Protein Assay (BCA method) using the manufacturer’s protocol (Thermo Scientific™ Pierce™ BCA™ Protein Assay) [44].

EVs labeled with Resovist^®^ were characterized by NTA and MRI. As described previously [31], size distribution and concentration of isolated EVs were measured by NTA using a NanoSight NS300 (Malvern Panalytical, Malvern, UK) equipped with a 488 nm laser and a syringe pump system with a pump speed of 20 µL/s. NTA 3.4 software (Malvern Panalytical) was used to acquire and analyze sample videos.

Three phantoms of EVs were prepared for analysis in MR imaging, (i) one with EVs isolated from cells labeled with 200 μgFe/mL of Resovist^®^, (ii) one with EVs isolated from unlabeled cells and used as a control (EVs CTRL), while a third sample contained a 1:10 dilution of (i) sample. The transverse relaxation time T_2_ of EVs solutions was measured by acquiring a T_2_ map MSME sequence, with the following parameters: TR = 10826 ms, 25 equally spaced echoes, TE min = 50 ms, TE max = 1250 ms, FOV = 5 × 5 cm^2^, MTX = 192 × 192, slice thickness = 1.5 mm, N slice = 6, and NEX = 1. A T_1_-weighted FLASH sequence was acquired with TR = 1000 ms, TE = 15 ms, flip angle = 15°, FOV = 5.3 × 3.9 cm^2^, MTX = 256 × 192, slice thickness = 0.5 mm, N slic e = 15, and NEX = 10.

## 5. Conclusions

This study underscores the importance of selecting appropriate NPs for labeling cells. Resovist^®^, with its larger core size and higher transverse relaxivity, has demonstrated superior performance in imaging contrast and detectability compared to USPIO. Additionally, Resovist^®^ demonstrates less cell toxicity and more efficient internalization within cells. These findings offer valuable insights for ongoing investigations into the therapeutic potential of indirectly labeled EVs derived from ADAS. The imaging of labeled cells and EVs holds promise for expediting the clinical translation of EV therapies, enhancing the precision and accuracy of regenerative nanomedicine. Future studies should focus on in vivo applications to validate these findings and explore the therapeutic potential of labeled EVs in clinical setting.

## Figures and Tables

**Figure 1 ijms-25-09701-f001:**
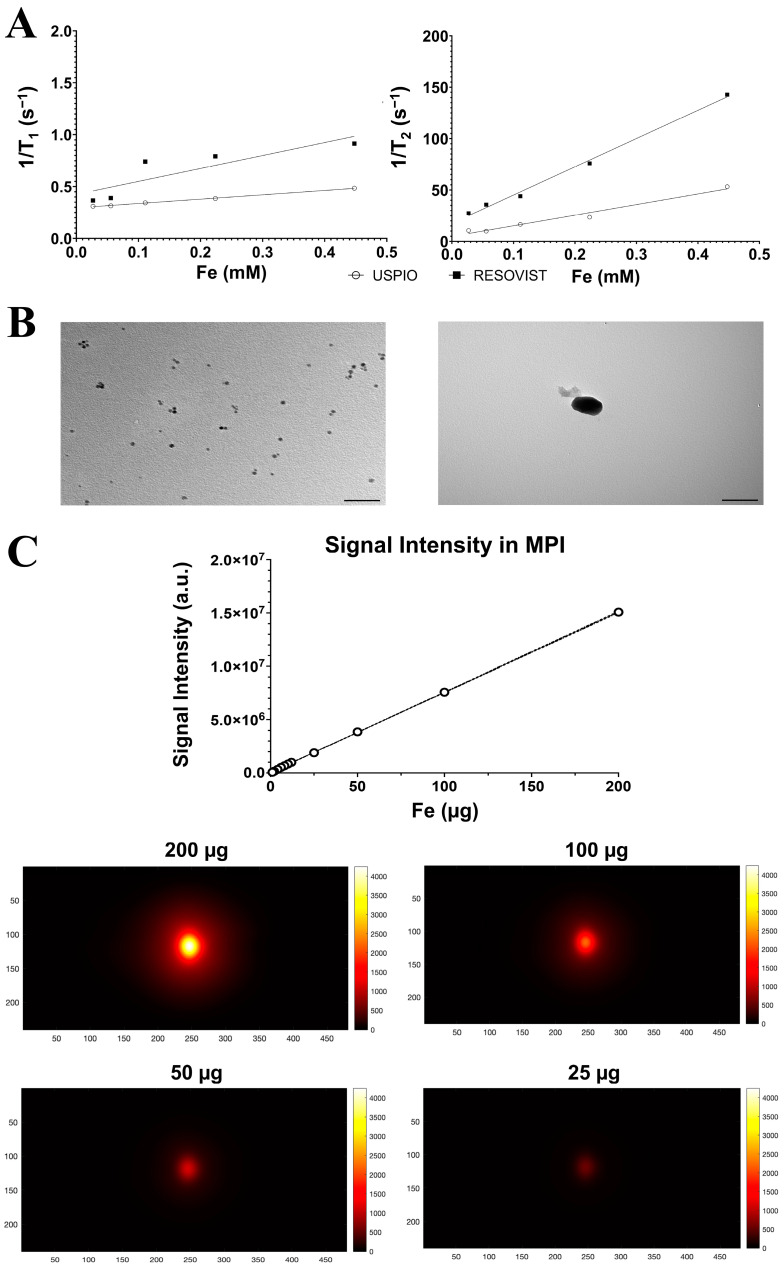
Physical–chemical characterization of nanoparticles. (**A**) Plot of 1/T_1_ and 1/T_2_ relaxation rates of USPIO and Resovist^®^ in Agar 2%; (**B**) Transmission electron microscopy (TEM) images of USPIO (left) and Resovist^®^ (right) nanoparticles. Scale bars: 50 nm; (**C**) MP images of different concentrations of Resovist^®^ at a field strength of 3.055 T/m. In magnetic particle images, the horizontal and vertical axis represent the number of pixels. Images are 250 × 500 pixels, and each pixel corresponds to 0.24 × 0.24 mm^2^. Signal intensity is expressed in arbitrary units.

**Figure 2 ijms-25-09701-f002:**
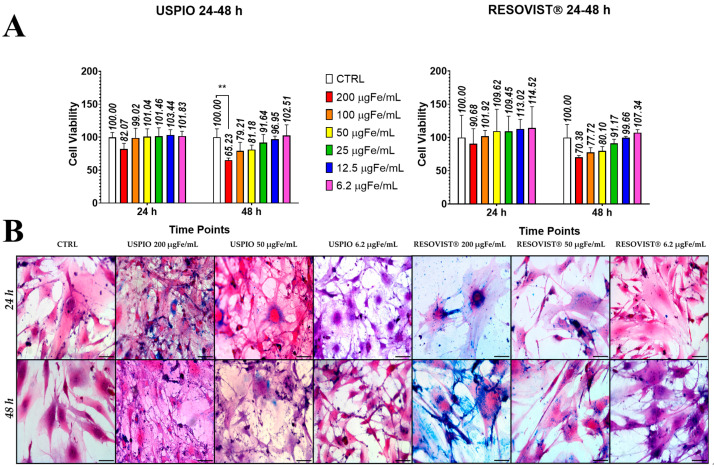
Internalization of nanoparticles into adipose tissue-derived adult stem (ADAS) cells. (**A**) MTT assay in ADAS cells shows that both USPIO and Resovist^®^ are safe for cell viability at concentrations up to 200 μgFe/mL and incubation time up to 48 h. USPIO at concentration of 200 μgFe/mL at 48 h showed a statistically significant difference in viability compared to control ADAS (CTRL), ** *p* < 0.01. Error bars represent SEM; (**B**) Optical microscopy images of Prussian Blue-stained ADAS labeled with USPIO and Resovist^®^ (20× Magnification and scale bar 50 µm) after 24 h (first row) and 48 h of incubation time (second row).

**Figure 3 ijms-25-09701-f003:**
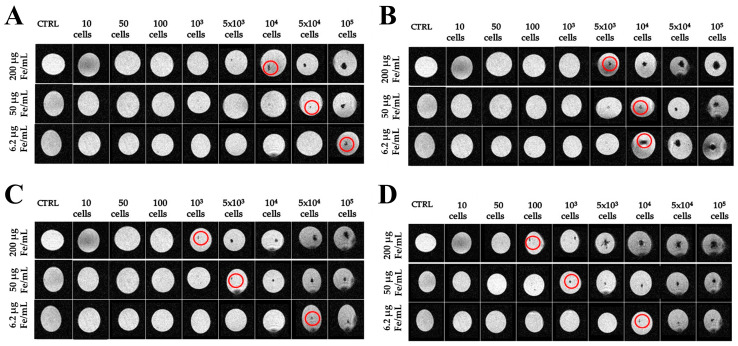
Magnetic resonance imaging (MRI) of agarose gel phantoms containing different amounts of labeled cells. Red circles indicate the hypointense spots attributable to the presence of nanoparticles inside the cells in fast low-angle shot (FLASH) sequence. (**A**) MRI phantom of ADAS cells labeled with USPIO for 24 h; (**B**) MRI phantom of ADAS cells labeled with USPIO for 48 h; (**C**) MRI phantom of ADAS labeled with Resovist^®^ for 24 h; (**D**) MRI phantom of ADAS cells labeled with Resovist^®^ for 48 h. MRI images were acquired using tubes of 1 cm diameter.

**Figure 4 ijms-25-09701-f004:**
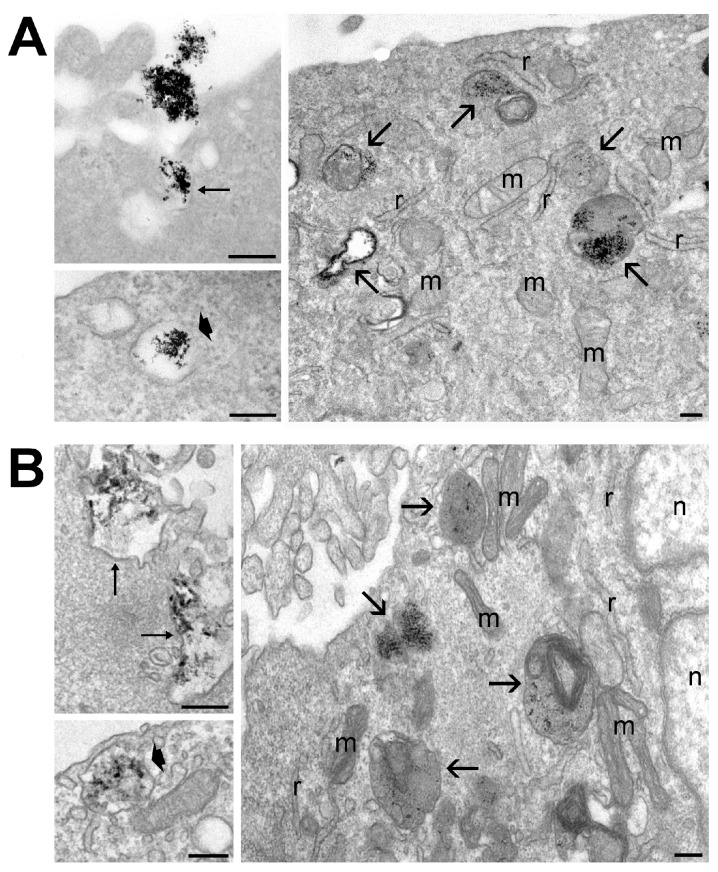
Transmission electron micrographs of ADAS cells treated with USPIO (**A**) and Resovist^®^ (**B**) for 24 h. Both nanoparticles enter the cells by endocytosis (thin arrows indicate cell surface invaginations containing nanoparticulate). In the cytoplasm, nanoparticles occur inside vacuoles (thick arrows) located just beneath the cell surface or inside heterogeneous vacuolar structures (arrows). Note the good preservation of cell organelles: m, mitochondria; r, endoplasmic reticulum; n, nucleus. Bars: 200 nm.

**Figure 5 ijms-25-09701-f005:**
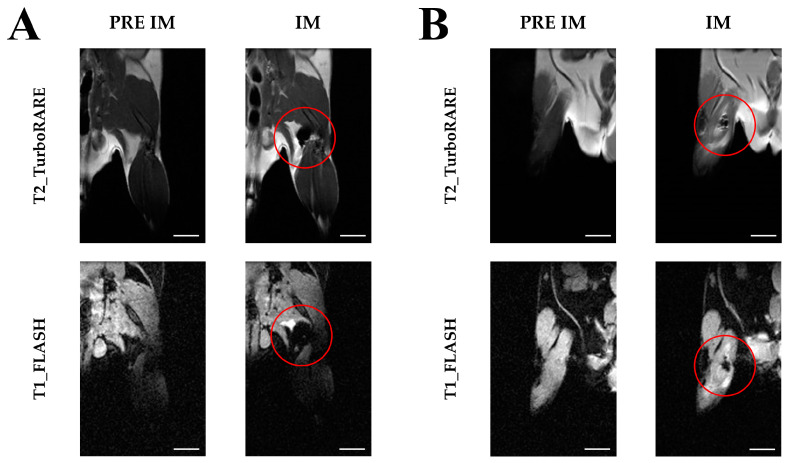
Imaging with magnetic resonance of intramuscular injection of ADAS labeled with USPIO and Resovist^®^. (**A**) Pre- and post-intramuscolar injecton of 5 × 10^3^ ADAS labeled with USPIO 200 µgFe/mL; (**B**) Pre- and post-intramuscolar injection of 5 × 10^3^ ADAS labeled with Resovist^®^ 200 µgFe/mL. Pre IM: pre-intramuscular injection; IM: after intramuscular injection. The red circles highlight areas where cells labeled with nanoparticles are present. Scale bar 5 mm.

**Figure 6 ijms-25-09701-f006:**
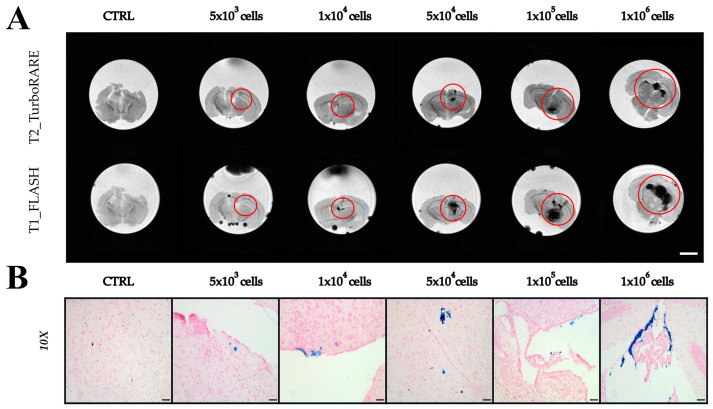
Imaging on ex vivo brain after injection of ADAS cells labeled with Resovist^®^. (**A**) MRI T2_TurboRARE and T1_FLASH sequence images (scale bar 4 mm); (**B**) Optical microscopy images of Prussian Blue staining (10× Magnification, scale bar 50 µm).

**Figure 7 ijms-25-09701-f007:**
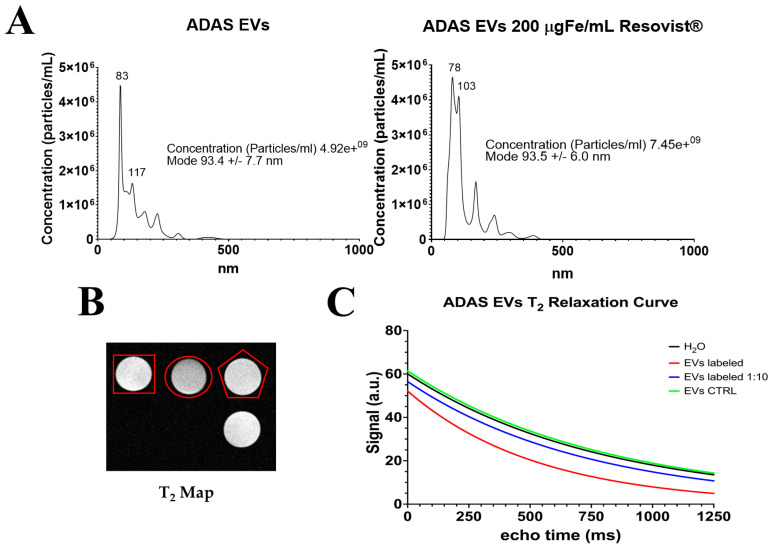
Extracellular vesicles (EVs). (**A**) Nanoparticle tracking analysis (NTA) of unlabeled EVs (on the left) and EVs labeled with 200 µgFe/mL of Resovist^®^ (on the right); (**B**) Representative MRI T_2_ Map on unlabeled EVs (red square), EVs labeled with 200 µgFe/mL of Resovist^®^ (red circle), a solution 1:10 of labeled EVs with 200 µgFe/mL of Resovist^®^ (red pentagon), and water not marked; (**C**) T_2_ relaxation curve of water (black), EVs labeled with 200 µgFe/mL of Resovist^®^ (red), EVs labeled 1:10 solution (blue), and EVs unlabeled CTRL (green).

**Table 1 ijms-25-09701-t001:** Physical–chemical characterization of nanoparticles.

	USPIO in Distilled Water	USPIO in Agar Gel 2%	Resovist^®^ in Distilled Water	Resovist^®^ in Agar Gel 2%
Size on Light Scattering (Size Peak nm ± SDev)	9.5 ± 2.01		69.18 ± 1.56	
Zeta Potential (mV ± ZDev)	−23.6 ± 1.7		−31.8 ± 7.12	
Size on Transmission Electron Microscopy (nm ± SDev)	5.0± 0.01		70 ± 0.2	
Transverse relaxivity (mM^−1^ s^−1^)	Not measured *	103.3 ± 2.122	176.6 ± 1.868	274.9 ± 4.108
Longitudinal relaxivity (mM^−1^ s^−1^)	Not measured *	0.4174 ± 0.018	1.736 ± 0.042	1.072 ± 0.07
Ratio r_2_/r_1_		247.484	101.73	256.436

* USPIO relaxivity in distilled water was not measured because, in these conditions, the nanoparticles tended to aggregate.

## Data Availability

The original contributions presented in the study are included in the article, further inquiries can be directed to the corresponding author.

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
