# Peer review of "Comparison between USPIOs and SPIOs for Multimodal Imaging of Extracellular Vesicles Extracted from Adipose Tissue-Derived Adult Stem Cells"

_ijms, 2024, doi:10.3390/ijms25179701_

Round 1
Reviewer 1 Report
Comments and Suggestions for Authors
This study investigates the efficacy of two different superparamagnetic iron oxide nanoparticles (SPIO and USPIO) for non-invasive adipose tissue-derived adult stem cells (ADAS) tracking via MRI/MPI and indirect labeling of EVs.
This work stands out due to its thoroughness and coherence, with well-constructed arguments and consistent results supported by high-quality images. The physicochemical characterization of the nanoparticles is meticulously performed using DLS, TEM, MRI, and MPI. ADAS labeling with these nanoparticles is evaluated for viability, and the detection of labeled cells is confirmed trough different techniques. The findings indicate that Resovist, due to its bigger nucleus and than its higher transversal relaxivity, provides superior sensitivity for MRI detection compared to USPIO. This study identifies the optimal concentration for ADAS labeling, and preliminary results suggest that EVs retain the nanoparticles, facilitating efficient and multimodal detection. Overall, the paper makes significant contributions to the field and presents promising avenues for future research.
Author Response
We thank you very much for taking the time to review this manuscript and for the appreciation of the manuscript.
Reviewer 2 Report
Comments and Suggestions for Authors
Capuzzo, et. al., produced a manuscript that details their comparison of ultra-small SPIONs vs. larger, clinically-approved SPIONs in how they can be used to label and track extracellular vesicles from adipose tissue-derived stem cells. This work follows a previous publication from the group that only examined the ultra-small SPIONs; this new work shows potential benefits of the larger, clinically-approved nanoparticles. There are some broad considerations about the choice of Resovist; while often used as a benchmark for MRI studies (and previously receiving clinical approval), its global availability and clinical use has been sporadic in the last 15 years. This doesn’t affect the science, but rather the potential impact. Overall, the manuscript can benefit from more comprehensive explanations for some of its reported data. Please see specific notes below:
-Authors should be consistent throughout with the format of T1 and T2 vs. T1 and T2 (e.g., subscript or not)
-Table 1: Why is there no relaxivity data for the USPIOs in DI water?
-Figure 1A: For the graph on the left (1/T1 vs. [Fe]), the Resovist data is fit to a linear function despite the data not supporting this choice of fit. Do the authors have a reason for fitting to this line, or would a different mathematical fit be a more appropriate choice for this data? Is there any scientific reason to choose a different fitting function? Also, the font size of the axis labels are inconsistent between the left and right.
-Figure 1B: These two TEM images are not terribly helpful. For the USPIO particles, it is too zoomed-out to see anything interesting. Perhaps a high-resolution image of a single particle as an inset, set against the more zoomed-out perspective would be more helpful. For the Resovist particle, the paper claims that it should be ~70 nm in size, but it looks smaller than the 50 nm scale bar. I think the manuscript should feature TEM images of the particles, but maybe replace these with more helpful images.
-Figure 1C: For the MP images on the right, the labels are too small to read. I assume x and y are distances, but they are not labelled as such (and if they were, I couldn’t read the values) and the labels of the intensity scale is also too small to read. I think that, overall, Figure 1 would benefit from some fine-tuning in terms of size/format and consistency in presentation.
-Figure 3: I’m a little confused by this figure, in that the concentration effect of iron doesn’t seem to correlate with the cell number. For example, Figure 3D: when the SPION concentration is reduced from 200 mg/mL to 6.2 mg/mL—a reduction of ~32x, it doesn’t require 32x more cells to be viewed—it requires 100x more cells. In the same figure, reducing from 200 mg/mL to 50 mg/mL requires 10x more cells instead of 4x more cells. In any case, I’m sure that the data ‘is what it is’, but a more comprehensive explanation of why such relationships are observed would be helpful to the reader.
-Figure 6: I question the impact of Figure 6 and fail to see the benefit of imaging the cells in excised mouse brains. If this was in vivo, then it’s a different story. I worry about cells leaking out from the injection site. I just don’t really see the point of this figure; unless the authors have a comprehensive justification for including it, then I would suggest removing it.
-Line 442: Should ‘underling’ instead be ‘underlying’?
-Line 478: To be fair, the difference in cell viability at 48 hrs is fairly similar between USPIO and Resovist. Yes, USPIO is worse at the highest concentration—but not by much. There are other concentrations where USPIO-tagged cells are more viable than the Resovist at 48 hrs. I would be cautious with language that may bias the reader.
-Line 506: The authors state that 72 hrs is an inappropriate choice of timescale for extracellular vesicle labelling without any context; could the authors provide a citation to support this statement, or additional information on how they arrived at this conclusion?
-Line 532: I’m curious as to why a FLASH imaging sequence would work better here than RARE; the manuscript would benefit from more information on why the low tipping angle sequence is preferred under these conditions compared to the T2-weighted sequence (since SPIONs function at relaxation agents).
-Line 546: There is some concern about the relative size between the 70 nm Resovist particles and the expected size of the extracellular vesicles. While the authors measured EV sizes of ~93 nm, if the SPION takes up most of that volume, are the EVs going to behave the way that they would without the particles? If the main goal is to track EVs as they provide therapy, would they still provide that therapy (or even move the same way) when they have a relatively large SPION inside? While the behavior of the EVs when loaded with a SPION is outside the scope of this work, it does question the potential future impact of this work.
-The work would be considered more robust if the authors would include more data in the Supplementary file, such as DLS histograms, additional TEM images, and other supporting data that was collected and discussed, but does not otherwise fit into the main manuscript in the form of a figure.
-There is a significant number of self-citations in the References section; my tally is that 24 out of 50 references are self-citations from various individuals in the author list. Please ensure that all references are appropriate and applicable; this self-citation rate of 48% is outside of common scientific publishing norms—especially for an original research publication.
Author Response
Reviewer N°2
We thank you very much for the time and effort you have dedicated to reviewing our manuscript. Please find the detailed responses below and the corresponding revisions/corrections highlighted in yellow in the re-submitted files.
Comments 1: Authors should be consistent throughout with the format of T1 and T2 vs. T1 and T2 (e.g., subscript or not)
Response 1: We would like to thank you for your comment; we have corrected the format of T1 and T2.
Comments 2: Table 1: Why is there no relaxivity data for the USPIOs in DI water?
Response 2: Thank you for pointing this to our attention. We did not measure USPIO relaxivity in pure water because the nanoparticles tended to aggregate under these conditions. This has been clarified in the note added to Table 1.
Comments 3: Figure 1A: For the graph on the left (1/T1 vs. [Fe]), the Resovist data is fit to a linear function despite the data not supporting this choice of fit. Do the authors have a reason for fitting to this line, or would a different mathematical fit be a more appropriate choice for this data? Is there any scientific reason to choose a different fitting function? Also, the font size of the axis labels are inconsistent between the left and right.
Response 3: Relaxation rate data typically follow a linear relationship between the concentration of a contrast agent (expressed in millimoles per liter, mM) and the reciprocal of the relaxation times, 1/T1 and 1/T2. This theoretical relationship implies that as the concentration of the contrast agent increases, the reciprocal of the relaxation times increases linearly. In our specific case, we chose a linear fit to maintain consistency with this well-established theoretical model and to facilitate the interpretation of the results in terms of relaxivity. Thank you for pointing out the error in the font size; we have since corrected it.
Comments 4: These two TEM images are not terribly helpful. For the USPIO particles, it is too zoomed-out to see anything interesting. Perhaps a high-resolution image of a single particle as an inset, set against the more zoomed-out perspective would be more helpful. For the Resovist particle, the paper claims that it should be ~70 nm in size, but it looks smaller than the 50 nm scale bar. I think the manuscript should feature TEM images of the particles, but maybe replace these with more helpful images.
Response 4: We agree with the reviewer that the TEM images could be improved by acquiring them at higher resolution. We have, therefore, acquired additional high-resolution TEM images, which have been included in the supplementary materials. We have also added the following sentence to the text: “To better illustrate the structure of the nanoparticles used, high-resolution TEM images are included in the supplementary materials (see Figure S2 and Figure S3).” in paragraph 2.1, lines 141-142.
Comments 5: Figure 1C: For the MP images on the right, the labels are too small to read. I assume x and y are distances, but they are not labelled as such (and if they were, I couldn’t read the values) and the labels of the intensity scale is also too small to read. I think that, overall, Figure 1 would benefit from some fine-tuning in terms of size/format and consistency in presentation.
Response 5: Thank you for highlighting this issue. In these images, X and Y axes represent the number of pixels. The images are 250 x 500 pixels in size, with each pixel corresponding to 0.24 x 0.24 mm2. This information has been added in the caption of Figure 1C. Moreover, we have revised Figure 1, adjusting its size, format, and overall consistency. The accompanying text has been updated from “On the left, a linear correlation between the Fe concentration and the signal intensity is evident. On the right, representative MPI images, showing signal intensities of Resovist®, are reported.” to “The upper part shows a linear correlation between the Fe concentration and the signal intensity. In the bottom part, representative MPI images showing signal intensities of Resovist® are reported” (paragraph 2.1, and lines 151-153) to accurately describe the new figure 1C. We hope that these changes meet your expectations and improve the overall presentation.
Comments 6: Figure 3: I’m a little confused by this figure, in that the concentration effect of iron doesn’t seem to correlate with the cell number. For example, Figure 3D: when the SPION concentration is reduced from 200 mg/mL to 6.2 mg/mL—a reduction of ~32x, it doesn’t require 32x more cells to be viewed—it requires 100x more cells. In the same figure, reducing from 200 mg/mL to 50 mg/mL requires 10x more cells instead of 4x more cells. In any case, I’m sure that the data ‘is what it is’, but a more comprehensive explanation of why such relationships are observed would be helpful to the reader.
Response 6: We thank the reviewer for raising this point, which has prompted us to clarify our results. First, we would like to clarify that the concentrations discussed refer to the concentration of nanoparticles in the cell culture medium. While it is true that an increase in nanoparticle concentration in the culture medium will lead to a corresponding increase in the amount of iron internalized by the cells, this relationship is not guaranteed to be linear. The second aspect to consider is that the MRI technique used in this context is not quantitative. This is a well-known limitation of the method, as reported in the literature (https://doi.org/10.1148/radiol.2018180449).
We have added the following sentence to the results to clarify this point in paragraph 2.2, lines 222-232: “Figure 3 shows that the relationship between the concentration of Fe added to the culture medium and the minimum number of cells detected by MRI is not linear. For example, when the SPION concentration is reduced from 200 μgFe/mL to 6.2 μgFe/mL —a reduction of approximately 32 times—it requires 100 times more cells to be detected. Similarly, reducing the concentration from 200 μgFe/mL to 50 μgFe/mL requires 10 times more cells instead of the expected 4 times more if the relationship was linear. These findings can be explained by the fact that, while increasing the concentration of NPs in the culture medium does lead to an increase in the amount of iron internalized by the cells, this increase is not necessarily linear. Moreover, MRI technique used in this context is not quantitative. This is a well-known limitation of the method, as reported in the literature (29).”
Comments 7: Figure 6: I question the impact of Figure 6 and fail to see the benefit of imaging the cells in excised mouse brains. If this was in vivo, then it’s a different story. I worry about cells leaking out from the injection site. I just don’t really see the point of this figure; unless the authors have a comprehensive justification for including it, then I would suggest removing it.
Response 7: We understand the concern raised by the reviewer regarding the impact of Figure 6 and agree that if the experiment had been conducted in vivo, it would have been significantly more meaningful. However, we believe that visualizing the cells in the brains of excised mice is important for our study, as our future goal will be to demonstrate that the target of the therapy is indeed the brain tissue. Confirming that the cells can be detected in the brain, even ex vivo, provides us with crucial insight into the feasibility of conducting in vivo experiments. Additionally, we would like to clarify that the administered volume was 10 µL, which is relatively substantial for a mouse brain. This may account for the leakage of cells from the injection site, a phenomenon that, while not ideal, is understandable given the small size of the brain and the high volume injected. While we cannot definitively state that 10 µL is excessive compared to the brain's volume, we believe it is significant in this context. Thus, Figure 6 serves to demonstrate that despite these technical challenges, the cells are still present in the target tissue, confirming the relevance of our approach. In this context, we find the figure justified and useful for a better understanding of our study's results.
However, acknowledging the reviewer's point, the following text has been added to the manuscript in paragraph 2.3, lines 291-299: “While a significant limitation of this experiment is that it was conducted ex vivo, we believe that visualizing cells in excised mouse brains remains crucial, as the target organ of the proposed therapeutic approach is brain tissue. Additionally, we would like to clarify that the administered volume was 10 μL, which is relatively large for a mouse brain. This may account for the observed cell leakage from the injection site, a phenomenon that, although not ideal, is understandable given the high injection volume. Nevertheless, Figure 6 demonstrates that despite these technical challenges, cells are detectable within the target tissue, underscoring the relevance of our approach.”.
Comments 8: Line 442: Should ‘underling’ instead be ‘underlying’?
Response 8: We thank the reviewer for pointing this out; it was a typographical error, which has since been corrected (paragraph 3,line 353).
Comments 9: Line 478: To be fair, the difference in cell viability at 48 hrs is fairly similar between USPIO and Resovist. Yes, USPIO is worse at the highest concentration—but not by much. There are other concentrations where USPIO-tagged cells are more viable than the Resovist at 48 hrs. I would be cautious with language that may bias the reader.
Response 9: We appreciate the reviewer for bringing up this issue. We modified the original text from: “However, after 48 h of incubation, a statistically significant difference (p<0.01) was observed between ADAS labeled with USPIO at a concentration of 200 µgFe/mL and CTRL.” to the new version at line 180-182, paragraph 2.2: “However, after 48 h of incubation, a small, although statistically significant, difference (p<0.01) was observed between ADAS labeled with USPIO at a concentration of 200 µgFe/mL and CTRL.” Additionally, we updated a sentence in the discussion from: “In the present study, however, the MTT assay revealed a significant difference between the control group and the USPIO-labeled cells at 48 h, indicating lower viability with USPIO compared to Resovist®.” to the new version at line 387-388, paragraph 3: “In the present study, however, the MTT assay revealed a small difference between the control group and the USPIO-labeled cells at 48 h”.
Comments 10: Line 506: The authors state that 72 hrs is an inappropriate choice of timescale for extracellular vesicle labelling without any context; could the authors provide a citation to support this statement, or additional information on how they arrived at this conclusion?
Response 10: The sentence in paragraph 4, page 14, lines 497-498 of original manuscript: “We opted not to investigate detectability in MRI at 72 h, as this labeling time point is not suitable for EVs labelling.” has been deleted.
Comments 11: Line 532: I’m curious as to why a FLASH imaging sequence would work better here than RARE; the manuscript would benefit from more information on why the low tipping angle sequence is preferred under these conditions compared to the T2-weighted sequence (since SPIONs function at relaxation agents).
Response 11: We thank the reviewer for rising the attention to this point. We decided to adopt a FLASH sequence for two reasons: first of all, a gradient-echo based sequences with a low flip angle (as is the FLASH) combined with a sufficiently long echo time (as in our case), is highly sensitive to changes in magnetic susceptibility thus enhancing the contrast in the image and allowing to properly identify the detection limit of the labelled cells/EVs; second (less important), a low flip angle FLASH sequence reduces the influence of T1 relaxation on the image, minimizing the T1 contrast component. This allows the imaging to focus more on the T2-related contrast, which is desirable for evaluating the effects of a T2-relaxing agent. We modified the text accordingly in paragraph 4.2, lines 541-542: “To enhance the sensitivity of image contrast to changes in local magnetic susceptibility,”.
Comments 12: Line 546: There is some concern about the relative size between the 70 nm Resovist particles and the expected size of the extracellular vesicles. While the authors measured EV sizes of ~93 nm, if the SPION takes up most of that volume, are the EVs going to behave the way that they would without the particles? If the main goal is to track EVs as they provide therapy, would they still provide that therapy (or even move the same way) when they have a relatively large SPION inside? While the behavior of the EVs when loaded with a SPION is outside the scope of this work, it does question the potential future impact of this work.
Response 12: To answer this question, the following sentence has been added to paragraph 3, lines 460-463: “This result is somewhat surprising since the 70 nm Resovist® particles have been, at least at a certain extent, incorporated into EVs. It remains to be elucidated whether this result can be due to modifications of EVs that could in principle affect the therapeutic efficacy.”
Comments 13: The work would be considered more robust if the authors would include more data in the Supplementary file, such as DLS histograms, additional TEM images, and other supporting data that was collected and discussed, but does not otherwise fit into the main manuscript in the form of a figure.
Response 13: Thank you for bringing this to our attention. In response, we have added DLS histograms of both nanoparticles to the Supplementary file (Figure S1). Additionally, we have acquired new TEM images of the nanoparticles to provide readers with a more comprehensive view (Figure S2 and Figure S3).
Comments 14: There is a significant number of self-citations in the References section; my tally is that 24 out of 50 references are self-citations from various individuals in the author list. Please ensure that all references are appropriate and applicable; this self-citation rate of 48% is outside of common scientific publishing norms—especially for an original research publication.
Response 14: Thank you for your observation. We have reviewed and reduced the number of self-citations. We hope that the revised list is now more in line with common scientific publishing norms.
Round 2
Reviewer 2 Report
Comments and Suggestions for Authors
Thank you for comprehensively addressing all prior comments; no additional comments are needed.